# Injuries in Novice Participants during an Eight-Week Start up CrossFit Program—A Prospective Cohort Study

**DOI:** 10.3390/sports8020021

**Published:** 2020-02-13

**Authors:** Rasmus Tolstrup Larsen, Andreas Lund Hessner, Lasse Ishøi, Henning Langberg, Jan Christensen

**Affiliations:** 1Department of Public Health, Section of Social Medicine, University of Copenhagen, 1123 Copenhagen K, Denmark; hessnerfysioterapi@gmail.com; 2Department of Orthopedic Surgery, Sports Orthopedic Research Center—Copenhagen (SORC-C), Copenhagen University Hospital, Amager-Hvidovre, 2650 Hvidovre, Denmark; lasse.ishoei@regionh.dk; 3Department of Public Health, Section for Health Services Research, University of Copenhagen, 1123 Copenhagen K, Denmark; henninglangberg@gmail.com; 4Department of Occupational and Physiotherapy, Copenhagen University Hospital, Rigshospitalet, 2100 Copenhagen Ø, Denmark; fysjan@gmail.com

**Keywords:** CrossFit, high-intensity functional training, injury rate, incidence, sport, exercise, training

## Abstract

Background: Previously published studies have reported injury rates ranging from 0.74 to 3.3 per 1000 h of exposure in CrossFit participants. However, the existing body of evidence is mainly based on experienced participants; therefore, the injury incidence and injury rate within novice CrossFit participants remains relatively unknown. The aim of this study wasto investigate the injury incidence and injury rate among novice participants in an eight-week CrossFit program. Methods: This survey-based prospective cohort study included CrossFit Copenhagen’s novice members who began an eight-week, free-of-charge membership period. A questionnaire was distributed at baseline and at eight-week follow-up. Information about exposure was retrieved through the online booking system. Injury incidence, defined as proportion of participants who sustained an injury, and injury rates per 1000 h of exposure were calculated. Results: Among the 168 included participants, a total of 28 injuries (14.9%) were reported. The number of injured participants and total exposure time resulted in an injury rate per 1000 h of exposure of 9.5. Conclusions: Compared to the existing body of evidence, the findings in this study indicate that the risk of injuries is higher among novice participants than among experienced CrossFit participants.

## 1. Introduction

CrossFit is a type of high-intensity functional training and is one of the fastest growing fitness programs in the fitness industry [1,2]. CrossFit aims to increase aerobic capacity and whole body muscle strength through a fitness program described as constantly varied functional movements performed at high intensity [3]. CrossFit and high-intensity functional training arereceiving a lot of criticism about the safety of the programs [4,5]. Several mechanisms for the presumed high number of injures have been discussed, including performing technical movements under high intensity close to exertion and even performing harmful movements [4,5]. However, these possible mechanisms are still undocumented, and a growing body of evidence concludes that the injury rates of CrossFit participants arecomparable to the injury rates of other noncontact sports such as Olympic weightlifting and gymnastics [4,6,7,8,9,10,11,12,13,14,15,16,17,18]. Among the studies reporting an injury rate per 1000 h of exposure, the results seem to be relatively consistent with reported injury rates ranging from 0.74 to 3.3 per 1000 h of exposure [4,6,7,8,9,10,11,12,13,14,15,16,17,19]. However, the vast majority of studies have used retrospective study designs [4,6,7,8,9,10,11,13,14,15,16,17], and only one study [12] has been designed to prospectively investigate the IR (Injury Rate).

Today, the existing body of evidence is, however, mainly based on experienced CrossFit participants or CrossFit athletes; therefore, the injury incidence and injury rate within novice CrossFit participants remains relatively unknown, even though two studies suggest that CrossFit participants with less than six months of experience are at greater risk of being injured when participating in CrossFit [11,17]. Knowledge on risk and benefits among novice CrossFit participants is important information, especially because the sport of CrossFit is rapidly growing. However, before introducing preventive measures or even establishing mechanisms of injuries, the extent of injury problems in a given sport should be investigated, as suggested by the Van Macheleninjuryprevention model [20].

Hence, the main aim of this prospective observational study was to investigate the incidence, measured as proportion of subjects with injuries, and injury rate per 1000 h of exposure among novice participants in an eight-week, free-of-charge CrossFit program. Furthermore, we investigated if the injury rate was higher among participants who attended introduction classes compared to participants who did not attend introduction classes.

## 2. Materials and Methods

### 2.1. Study Design

This study wasa prospective cohort study, and reporting adhered to “The Strengthening the Reporting of Observational Studies in Epidemiology” (STROBE) statement: guidelines for reporting observational studies [21].

### 2.2. Setting

With 25 gyms located in the larger cities in Denmark, CrossFit Copenhagen is the largest Danish commercial provider of CrossFit. CrossFit Copenhagen organized “The CrossFit Experiment”, an eight-week, free-of-charge membership period form 1 March 2018 for people without CrossFit experience who were not members already. 

Weekly, 1500 one-hour CrossFit sessions were available to the members. All of the CrossFit sessions included an experienced CrossFit trainer. Voluntary introduction classes were scheduled daily with the aim of introducing basic and fundamental movements used in CrossFit. Furthermore, participants had access to opengym training sessions without supervision from trainers. 

### 2.3. Participants

The CrossFit Experimentwas advertised by CrossFit Copenhagen using social media. Participants had to be above 18 years of age and could not be current members of CrossFit Copenhagen. Participants who signed up for The CrossFit Experiment were given an eight-week, free-of-charge membership. No requirements on training sessions were imposed. Participants were encouraged to participate in three introductory classes before being recommended to participate in two to three group-based workouts per week. 

Specifically, to be eligible for inclusion in this study, an additional sign-up process was needed. An auto-generated e-mail was sent to the e-mail address provided by the participants in the sign-up process for The CrossFit Experiment. The e-mail explained the purpose of this study and provided the participants with a link for the baseline questionnaire. As we needed the e-mail information from the participants, only participants who opened the e-mail with the online survey were considered eligible for inclusion in this study.

### 2.4. Data Sources

The online survey platform SurveyXact (www.surveyxact.dk, Ramboell Management, Aarhus, Denmark) was used to distribute a baseline and end-point survey. The baseline questionnaire consisted of 25 items focusing on participant demographic and previous injuries. The questionnaire was electronically distributed to the participants’ e-mails at the start of The CrossFit Experiment, and in case of non-response and e-mail reminder was sent six days later. 

The end-point questionnaire was distributed at the end of the eight-week CrossFit Experiment period to those participants who completed the baseline questionnaire. In case of non-responders, e-mail reminders were sent after 6, 11 and 16 days. The questionnaire consisted of general questions about the experience of participating in The CrossFit Experiment and questions about exposure and injuries. 

### 2.5. Outcome Variables

#### 2.5.1. Exposure

Information about training history (exposure time) was collected through the operating system of CrossFit Copenhagen. Information related to open gym exposure (sessions and time) and other sport activities was retrieved through the end-point questionnaire.

#### 2.5.2. Registration of Injuries

To have a sensitive and specific investigation of injuries, the OSTRC (Oslo Sports Trauma Research Centre) Overuse Injury Questionnaire was used as a template for the questions [22]. The above-mentioned questionnaire was developed by Clarsen et al. to have a valid measurement tool in terms of registration of overuse injuries in sports [22]. Thus, injuries were investigated using the question “During the CrossFit Experiment have you had any problems (pain, soreness, stiffness or swelling) related to your CrossFit training?” that could be answered with “yes or no”, and if the response was “yes” the relevant body regions (shoulder, neck, elbow/wrist, lower back, hip, knee, ankle and other areas) should be indicated. The participants were asked how long the problem occurred and to what extent it affected their training (full participation without problems, full participation but with problems, reduced participation, no ability to participate).

Furthermore, separately for each body part the following information was registered: (1) whether the participant believed the problem was provoked by a previous problem or if it was a new problem, (2) whether the participant believed it was an acute or an overuse injury. 

An injury was defined when two criteria were present: (1) reporting a problem defined as having pain, soreness, stiffness or swelling in one or more body regions and (2) being affected by the problem to an extent that resulted in reduced participation inthe CrossFit training for at least seven days. 

#### 2.5.3. Injury Rate

The primary outcome of interest was injury rate per 1000 h of exposure. To calculate this, exposure (h of training) and information on injuries was used. The exposure hours were calculated by objective, system-registered classattendance information plus self-reported training in the open gym training facilities. 

#### 2.5.4. Risk Factors 

Baseline characteristics and information about training volume and training habits in the study period were investigated as possible risk factors for injuries. The following variables were used in the analysis: sex [15], age [11], body mass index [9,16], weekly alcohol intake, previous musculoskeletal problems [6], training volume [16,19], amount of self-training in open gym [16], higher attendance in other sports during the CrossFit program [9,11], participation in introduction classes [15,23], previous exercise habits and if the participant was following the physical activity recommendations.

#### 2.5.5. Rhabdomyolysis

Due to some previous reporting of exercise-induced rhabdomyolysis [10,24], the following question was used to investigate incidence of the condition: “Were you in the period you participated in the CrossFit program medically diagnosed with Rhabdomyolysis?” 

### 2.6. Bias

Potential reporting bias, including social desirability bias, was addressed by using objectively measured exposure time retrieved from the operating system of CrossFit Copenhagen. 

### 2.7. Study Size

Only the participants who answered the baseline and end challenge questionnaire were included in the injury analysis. 

### 2.8. Ethics

Handling of data was approved by the University of Copenhagen, case number 504-0023/16-3000, Journal-number SUND-2018-16. As this study only consisted of a survey investigation of participants who signed up for participation in The CrossFit Experiment before being informed about this study, an approval from the National Committee on Health Research Ethics was not necessary. Signing-up for participation and completion of the baseline questionnaire was considered as written consent for participation. 

### 2.9. Statistical Methods

Normal distribution of continuous data was assessed visually by quantile–quantile plots of the standardized residuals. Normally distributed continuous data weresummarized with means, standard deviations and 95% confidence intervals (95%CI). Data without normal distribution or ordinal data were presented with medians and interquartile ranges. Frequencies and percentage of total were used to present categorical data. Differences in demographical data between participants who were lost to follow up and participants who were included in the final analysis were analyzed with chi-squaredtest for categorical data, Kruskal–Wallis test for ordinal and nonnormally distributed interval data and t-tests with unequal variance for interval data. 

To investigate if reporting bias was a problem, a comparison between system-registered number of classes attended and the participants reported attendance was calculated using Student’s paired t-test. 

The injury rates per 1000 h of exposure were calculated for both number of injuries and number of injured participants using the formula below [25,26], where “events” are defined as injuries or injured participants and “n” are number of total participants:$$IR\;hours=\frac{Events}{n*mean\;hours\;of\;exposure}*1000$$

Univariate logistic Poisson regressions were used to estimate incidence rate ratios (IRRs) with a 95%CI related to the possible risk factors of sex, age, Body Mass Index (BMI), weekly alcohol intake, problems prior to participation, system-registered class attendance, self-reported training in open gym, self-reported participation in other sports, introduction class attendance, previous participation in sports and meeting physical activity recommendations. 

Explorative associations for participation in introduction classes were performed using standard logistic regression models with odds ratios (ORs). Explorative sensitivity analyses with a less sensitive definition of injuries were performed using the injury definition made by Weisenthal et al. [15] and used in several other studies [11,13,14], where an injury was defined as present when meeting one of the following criteria: (1) total removal from CrossFit training and other outside routine physical activities for >1 w, (2) modification of normal training activities in duration, intensity, or mode for >2 w, or (3) any physical complaint severe enough to warrant a visit to a health professional. 

StataCorp 2017 (Stata Statistical Software: Release 15. College Station, TX, StataCorp LLC) was used for all statistical analyses and visualizations. An alpha level of 0.05 was considered the threshold for statistical significance. 

## 3. Results

### 3.1. Participants 

The study flow is visually presented in Figure 1. 

Of the 439 participants who opened the email, 303 answered the baseline questionnaire and 168 participants answered both the baseline and the end-point questionnaire after eight weeks (response rate: 38.3%). Characteristics of the 135 participants who answered the baseline but not the end-point questionnaire and the 168 participants who answered both the baseline and the end-point questionnaire are listed in Table 1. Those included in the final analysis were, on average, 2.4 years older (26.8 ± 6.6 and 29.2 ± 7.9 respectively). No other significant differences between responders and dropouts were present. 

### 3.2. Exposure

The 168 participants attended a total of 2362 classes with a median and IQR attendance of 14 (7 to 21.5). In total, the 168 participants reported to use the open gym for 272.5 h in the eight weeks, with a median and IQR weekly training hours in open gym facilities of0 (0 to 0). In total, the participants performed (system-registered attendance and self-reported open gym hours) 2634.5 h of training. Participants reported attendance was not different from the system-registered number of classes attended (0.4 more classes, 95%CI (−1.4 to 2.2), *p* = 0.644). 

Forty-nine participants (29.2%) reported to attend all three introduction classes. Seventy-two participants (42.9%) reported to attend one or two introduction classes. Forty-seven participants (28.0%) reported to not attend any introduction classes. The median and IQR weekly participation in other sports was 0.8 h (0 to 2). 

### 3.3. Injury Incidence

In total, 22 participants reported 1injury (13.1% of the total number of participants), whereas 3 participants reported 2 injuries (1.8%), giving a total of 28 injuries among the 168 participants. 

The most frequent injured anatomical locations were lower back with seven injuries (equivalent to 25%), knee with six injuries (21.4%), elbow/hand with five injuries (17.9%), “other anatomical locations” with five injuries (17.9%), shoulder with two injuries (7.1%), neck with one injury (3.6%), hip with one injury (3.6%), and ankle with one injury (3.6%). Twenty-one injuries (75%) were reported to be acute, 21 injures (75%) were believed to be provoked by previous injuries; thus, seven (25%) of the injuries were believed to be new injuries. 

### 3.4. Injury Rate

The number of injured participants (25) and total exposure time (classes and open gym) equaled an injury rate per 1000 h of exposure of 9.5. The total injuries (28 injuries) and total exposure time equaled an injury rate per 1000 h of exposure of 10.6. The number of injuries not reported to be provoked by old injuries (7 injuries) and total exposure time equaled an injury rate per 1000 h of exposure of 2.66. 

### 3.5. Risk Factors

Table 2 reports the injury rate in relation to different risk factors. The injury rate was not significantly associated with attending one or two (*p* = 0.080) or all introduction classes (*p* = 0.165). None of the other risk factors were found to be significant. 

### 3.6. Rhabdomyolysis

One person reported to have been medically diagnosed with exercise-induced rhabdomyolysis.

### 3.7. Explorative Associations

Participants who reported to be physically active with moderate to high intensity for at least 30 min/d were less likely to participate in introduction classes (OR: 0.43, 95%CI: (0.21 to 0.88), *p* = 0.021). Furthermore, participants who reported to be physically active with high intensity twice a week of at least 20 min duration were less likely to participate in introduction classes (OR: 0.27, 95%CI: (0.12 to 0.61), *p* = 0.02).

### 3.8. Explorative Sensitivity Analysis

Using the definition of Weisenthal et al. [15], 21 (12.5%) participants were defined as having an injury. Of these, 18 (10.7%) were categorized as having one injury, and 3 (1.8%) were categorized as having two injuries, giving a total of 24 injuries. The 21 injured participantsand total exposure time (classes and open gym) equaled an injury rate per 1000 h of exposure of 8.0. The total injuries (24 injuries) and total exposure time equaled an injury rate per 1000 h of exposure of 9.1. 

## 4. Discussion

The main findings of this study include the injury rate per 1000 h of exposure of 9.5 for participants, and 10.6 for total injuries. We did not find any significant association between taking introduction classes and the IRR.

### 4.1. Results

#### 4.1.1. Incidence and Injury Rate 

We found a total injury incidence of 14.9% among novice CrossFit participants included in this study and exposed to eight weeks of CrossFit training. The injury incidence found in this present study is low compared to findings in other studies investigating injury incidence in CrossFit [11,15]. However, both of the above-mentioned studies are based on a longer study period and, thus, a longer exposure time. As a linear relationship between the incidence and the exposure time cannot be expected, a better way to compare and interpret findings across studies is to compare the injury rate per time period (injury rate per 1000 h of exposure).

In our sample of novice CrossFitters, we found an injury rate per 1000 h of exposure of 9.5. This result is different from the injury rates reported in the literature of experienced CrossFitters, ranging from 0.74 to 3.3 per 1000 h of exposure [4,6,7,8,9,10,11,12,13,14,15,16,17,19]. However, this was expected, as a previous study found the rate of injuries to be inversely proportional to years of experience and that participants with less than six months of experience reported the highest rates of injuries [17]. The study found that those participants with less than sixmonths of experience and lowest weekly participation had an injury rate per 1000 h of exposure of 3.90 [17]. Even though this is higher than the previously reported results [4,6,7,8,9,10,11,12,13,14,15,16,19], it is still lower than the injury rate per 1000 h of exposure of 9.5 found in this present study.

#### 4.1.2. Injury Location

We found the lower back (25%) to be the most commonly injured body region, followed by knees (21%), elbow/hands (18%), other anatomical locations (18%), shoulders (7%), neck (4%), hips (4%) and ankles (4%). The results in this study are consistent with the results from the prospective study by Moran et al. [12] who also reported the lower back (33%) to be the most commonly injured body region, followed by the knee (20%), wrist (13%), thigh (13%), shoulder (7%), elbow (7%) and foot (7%) [17]. In the retrospective cross-sectional study by Chachula et al., lower back injuries were also found to be the most prevalent [7]. However, 8 of 12 located observational studies, investigating the injury incidence in CrossFit an reporting injury distribution, found the shoulder to be the most prevalent location of injury [4,8,11,15,16,17,19,27]. In three of the located studies, shoulder injuries made up more at least one-third of all injuries, which is significantly lower than in this present study [8,17,19]. Between-study variance should be expected, and it can only be hypothesized if novice CrossFit participants are more commonly injured in the lower back, compared to more experienced participants. This is an area for future research to investigate. 

#### 4.1.3. Risk Factors

Our injury rate is exclusively based on novice CrossFit participants. In comparison with the research on experienced participants from other studies, it seems that the novice participants in our study were at greater risk of injury, which is in line with previously published results that suggest a higher injury rate among novice participants [11,17]. Within our study, we did not find any significant risk factors associated with injuries, but the statistical power to reject all risk factors, per se, is unavailable. With the reported incidence rate ratio of 3.04 (0.88 to 10.60) for those who reported to participate in one or two introductory classes, compared to those who did not report to participate in any introductory classes, it seems that those who participated in introductory classes were more likely to report an injury during the study. The finding is insignificant, but still relevant. This was not expected, as previous studies have found beginner classes to protect against injuries [15,23]. However, this can also be easily explained by the finding from the explorative logistic regression model revealing that participants who reported to meet the physical activity recommendations prior to participation in this study were less likely to participate in introduction classes. Thus, those who participated in introduction classes could be expected to have had a lower level of physical activity. In summary, this finding should increase the focus on introduction classes and the ones participating in these, as they are expected to have a lower level of habitual physical activity and could have an increased risk of injuries. 

#### 4.1.4. Exertional Rhabdomyolysis 

Several published studies investigating benefits or risks of CrossFit have highlighted the concern of health consequences to the popular training program, such as exertional rhabdomyolysis. In this study we report one case, where the participant reported to have been medically diagnosed with exercise-induced rhabdomyolysis. 

A retrospective cross-sectional study of patients presenting with rhabdomyolysis during a period of 12 months in 2013 and 2014 at two tertiary referral hospitals in Melbourne, Australia, reported that 12 of 34 cases of rhabdomyolysis wereexertional in origin. Of these 12 cases, 5were reported to be caused by CrossFit [24]. Another retrospective cohort study found that of 523 patients presented with injuries associated with CrossFit activities. Of these, 11 patients ultimately received a diagnosis ofrhabdomyolysis [10]. Several published observational studies investigating the injury incidence in CrossFit have not reported the incidence of rhabdomyolysis [11,12,15,16,17,19,23,27]. However, in a retrospective survey by Escalante, there was also one case of rhabdomyolysis reported in a total of 159 included participants [8]. In retrospective cross-sectional studies by Hak (2013), rhabdomyolysis incidence is presented, but no cases of rhabdomyolysis were found [4]. In light of the available evidence, one case of rhabdomyolysis in 168 participants does not seem unlikely. However, caution must be taken when concluding upon this finding, as one case will always hold a high degree of imprecision. To investigate this further, future research should include measures of rhabdomyolysis in CrossFit participants.

## 5. Methodological Considerations

### 5.1. Definition of Injury

Our original definition of an injury was made to be as specific and sensitive as possible. Our explorative sensitivity analysis, using the injury definition of Weisenthal et al. [15], where modification of training had to last for 14 days, showed that four of the previous categorized injuries were not defined as an injury, giving injury rates per 1000 h of exposure of 8.0 and 9.1 for injured participants and total injuries, respectively. The difference in injury rate per 1000 h of exposure between the two definitions was not large enough to affect our conclusion, but it merely highlights that self-reported injury rates are affected by the sensitivity and specificity of the criteria. 

This study was conducted using a prospective study design, including a baseline measurement to gain knowledge about participant characteristics and injury history before beginning the CrossFit program. However, as we only investigated injuries after the CrossFit program ended, the nature of our injury results is retrospective. Hence, we were not able to conduct any formal personal assessment of the injuries and must rely on the information provided by the participants. Potentially, this may have resulted in less injuries being reported due to recall bias. However, thestudy and thus the recall period was limited to eight weeks, and in terms of participant-reported attendance, the participants were able to recall this compared to what was registered by the system.

### 5.2. Inclusion and Response Rate

Given the response rate (38%), it is possible that the non-responders showed discrepancies in injury rate and, thus, could change our results. We know that 135 participants who were lost to follow up, on average, were 2.4 years older and were less likely to have completed higher education, but they also did not differ in the other variables compared to the participants included in the final analysis. We cannot know if some of the participants lost to follow up chose not to answer the endpoint questionnaire because they had stopped the CrossFit program, due to an injury, or that they did not complete the end-point questionnaire because they did not sustain an injury. Therefore, our results can both over- and underestimate injury rates.

### 5.3. General Limitations and Recommendations for Future Research

This study used an observational design to investigate injury rates among novice CrossFit participants. Given the nature of the design, conclusions about causality cannot be drawn, and the results merely describe the injury rates among the participants. The results of this study should, therefore, be used to guide future research instead of guiding clinical recommendations about preventive measures. Future steps for investigation include challenging or replicating our results, establishing etiology and mechanisms of injuries among novice participants and then investigate effectiveness of introduced preventive measures such as on-ramp programs, as suggested by the Van Mecheleninjuryprevention model [20].

## 6. Conclusions

Among the participants who completed both questionnaires, 28 injuries were reported giving an injury incidence on 14.9%. The injury rates per 1000 h of exposure for injured participants were 9.5 and 10.6 for total number of injuries. Like all of the associations between risk factors and injuries, the associations between taking introduction classes and being injured were not significant. The results from this prospective study of novice CrossFit participants in Denmark revealed a higher injury rate, compared to the injury rates from previously published studies including experienced CrossFit participants, ranging from 0.74 to 3.3 per 1000 h of exposure [4,6,7,8,9,10,11,12,13,14,15,16,17,19]. In light of our results, novice participants might benefit from increased attention from coaches, and future on-ramp programs should be evidence-based and, thus, investigated further. 

## Figures and Tables

**Figure 1 sports-08-00021-f001:**
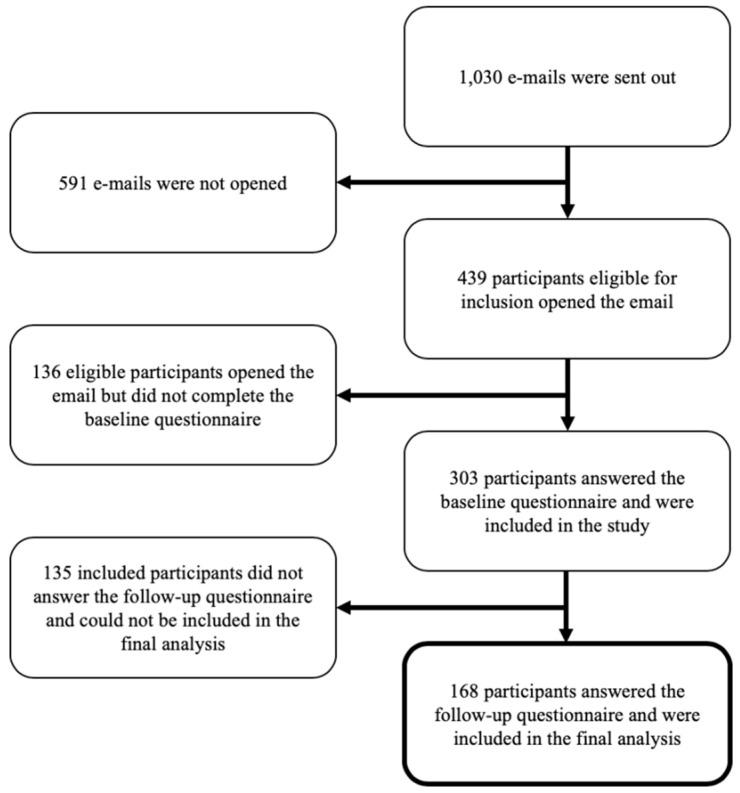
Participant flow-chart.

**Table 1 sports-08-00021-t001:** Participant demographics, *n* = 303.

	Lost to Follow up, *n* = 135	Included in Final Analysis, *n* = 168	*p*
**Age**, mean ± SD (95%CI)⊘	26.8 ± 6.6 (25.7 to 28.0)	29.2 ± 7.9 (28.0 to 30.4)	***p* = 0.006**
**Sex, male**, *n* (%)⨀	42 (31.1%)	51 (30.7%)	*p* = 0.888
**BMI**, mean ± SD (95%CI)⊘	24.5 ± 5.4 (23.5 to 25.4)	24.3 ± 2.9 (23.9 to 24.7)	*p* = 0.799
**Educational level**, *n* (%)⨀	9 (6.7%)	7 (4.2%)	***p* = 0.045**
Primary education	76 (56.3%)	71 (42.6%)	
Secondary education	36 (26.7%)	57 (33.4%)	
Tertiary	14 (10.4%)	32 (19.1%)	
Master’s, Doctoral or equivalent education	0 (0%)	1 (0.6%)	
“Do not know”			
**Currently studying**, *n* (%)⨀	68 (50.4%)	86 (51.2%)	*p* = 0.887
**Employed**, *n* (%)⨀	93 (68.9%)	120 (71.4%)	*p* = 0.631
**Currently smoking**, *n* (%)⨀	12 (8.9%)	12 (7.2%)	*p* = 0.576
**Units of alcohol per week**, median (IQR)⊗	2 (0.2 to 5)	2.5 (1 to 5)	*p* = 0.307
**Previous primary sport**, *n* (%)⨀			*p* = 0.052
CrossFit or functional training	1 (0.7%)	2 (1.2%)	
Cycling	4 (2.9%)	12 (7.1%)	
Dance	2 (1.5%)	2 (1.2%)	
Fitness	43 (31.9%)	34 (20.2%)	
Team sports (handball, soccer etc.)	10 (7.4%)	25 (14.9%)	
Gymnastics	6 (4.4%)	2 (1.2%)	
Running	19 (14.1%)	40 (23.8%)	
Strength training	14 (10.4%)	12 (7.1%)	
Swimming	2 (1.5%)	5 (3.0%)	
Yoga or Pilates	3 (2.2%)	5 (3.0%)	
Other sports	11 (8.2%)	12 (6.6%)	
I did not exercise previously	14 (10.4%)	15 (8.9%)	
Unanswered	6 (4.4%)	2 (1.8%)	
**Eating habits prior to participation**, *n* (%)⨀			*p* = 0.331
Did not think about eating habits	13 (9.6%)	16 (9.5%)	
Thought a little bit about eating habits	79 (58.5%)	107 (63.7%)	
Thought a lot about eating habits	37 (27.4%)	43 (25.6%)	
Unanswered	6 (4.4%)	2 (1.2%)	
**Physically active at least 30 min daily with moderate to high intensity**, *n* (%)⨀	68 (50.4%)	83 (49.4%)	*p* = 0.096
Unanswered	10 (7.4%)	4 (2.4%)	
**Physically active with high intensity twice a week of at least 20 min duration**, *n* (%)⨀	75 (55.6%)	99 (58.9%)	*p* = 0.117
Unanswered	10 (7.4%)	4 (2.4%)	
**Having pain, soreness, stiffness or swelling within the last two weeks in at least one body region prior to CrossFit**, *n* (%)⨀	70 (54.26%)	87 (52.4%)	*p* = 0.752

Abbreviations: SD: Standard deviation, BMI: Body mass index, IQR: Interquartile range. “Currently studying” and “Employed” were two different questions and could be answered independently fromeach other. ⨀ Between group difference analyzed with Chi^2^ test, ⊗ Between group difference analyzed with Kruskal–Wallis test, ⊘ Between group difference analyzed with a t-test using unequal variance. Significant between group differences are marked with bold. The currently studying and employed variable adds up to more than 100% as these outcomes were investigated separately; therefore, a participant could both be studying and be employed.

**Table 2 sports-08-00021-t002:** Univariate logistic Poisson regressions and incidence rate ratios from risk factors.

Risk Factor	Number of Participants	IRR (95%CI)	*P*
**Male**	*n* = 51	0.57 (0.22 to 1.53)	*p* = 0.266
Reference: female	*n* = 117		
**Higher age**	*n* = 168	0.97 (0.92 to 1.02)	*p* = 0.247
**Higher BMI**	*n* = 168	1.02 (0.89 to 1.16)	*p* = 0.818
**Higher number of weekly units of alcohol**	*n* = 168	1.05 (0.95 to 1.15)	*p* = 0.366
**Having pain, soreness, stiffness or swelling within the last two weeks prior to CrossFit**	*n* = 88	0.87 (0.40 to 1.97)	*p* = 0.768
Reference: no problems	*n* = 78		
**Higher number of system-registered CrossFit class attendance**	*n* = 164	0.96 (0.92 to 1.01)	*p* = 0.144
**Higher number of reported minutes of weekly Open Gym training**	*n* = 168	1.00 (1.00 to 1.00)	*p* = 0.811
**Higher number of minutes reported for participation in other sports**	*n* = 168	1.00 (1.00 to 1.00)	*p* = 0.509
**Participated in one or two introduction classes**	*n* = 47	3.04 (0.88 to 10.60)	*p* = 0.080
**Participated in all introduction classes**	*n* = 72	2.56 (0.68 to 9.64)	*p* = 0.165
Reference: did not participate in introduction classes	*n* = 47		
**Reported to participate in sports or exercise prior to CrossFit**	*n* = 152	1.09 (0.26 to 4.62)	*p* = 0.912
Reference: did not report to participate in sports or exercise	*n* = 15		
**Reported to meet one of the physical activity recommendations**	*n* = 60	1.72 (0.60 to 4.96)	*p* = 0.313
**Reported to meet both of the physical activity recommendations**	*n* = 61	1.39 (0.46 to 4.14)	*p* = 0.558
Reference: did not meet any of the physical activity recommendations	*n* = 47		

Abbreviations: 95%CI: 95% confidence interval, IRR: Incidencerate ratio, BMI: Body mass index, IQR: Interquartile range. Significant between-group differences are marked with bold. Reference values: 1.00.

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
