# Peer review of "Injuries in Novice Participants during an Eight-Week Start up CrossFit Program—A Prospective Cohort Study"

_sports, 2020, doi:10.3390/sports8020021_

Round 1

Reviewer 1 Report

The direction of research seems to be very necessary, considering more and more injuries in people training CrossFit. Awareness of the problem should penetrate to both participants and coaches as well as club owners.

Critical remarks.

Lack of information and literature references showing the reason why CrossFit has a large number of injuries. In my opinion, the main reason for injury in CrossFit is the fact that people exercise to the limit of their possibilities, which means that the correctness of the exercise performance decreases with the duration of the exercise. Lack of proper movement pattern during exercise causes that loads in the musculo-sceletal system increase, which in turn will have a great impact on injuries (both those mentioned by the authors arising in the short term, and those that may appear after years of doing this type of exercise ).

Considering the articles [10, 14] mentioned by the authors where the same studies as those described in the reviewed article were carried out, an attempt to investigate the reasons for the injury and referring the results to this information would significantly enrich the scientific value of the publication.

Reviewer 2 Report

In opinion of this reviewer, this study is original and is a high contribution to this type of exercise programme, some minor details along the text should be adressed:

p.1, line 41. Check spaces between word ().

Font size of sub sections should be adjusted (in section 2., these titles seems very big compared with main title).

6, lines 203-211. The use of number and letter in the text along the paragraph “Injury incidence” is confusing. Please follow journal / Vancouver rules to refer at numbers along the text. 8, line 253-236. This phrase should be rewritten as no clear result is showed. 8, line 242. This “results” subheading should be removed. 8, line 249. Error in “stu rdies”.

Reviewer 3 Report

The study investigates the injury incidence and injury rate among novice participants who participated in an eight-week CrossFit program. The authors concluded that in comparison of the existing body of evidence, the findings in this study indicate that the risk of injuries is higher among participants than among experienced CrossFit participants.

The results of this research are in my opinion of relevance to the field of sports science and would fit the scope of the journal. There are, however, minor concerns which should be addressed.

I would suggest to specify the most frequent injuries among novice participants who participated in the CrossFit program

Page 1. lines 29-31: Results: Among the 168 included participants, a total of 28 injuries (14.9%) were reported. The number of injured participants and total exposure time resulted in an injury rate per 1000 hours of exposure on 9.5.

Though the authors included “Methodological considerations” related to definition of injury and inclusion and response rate, they should discuss more relevant limitations of this study.

The authors should also present the practical applications of their findings with respect to novice CrossFit participants that are not currently addressed in the literature. 
